# Identification of a Novel de Novo Variant in the *SYT2* Gene Causing a Rare Type of Distal Hereditary Motor Neuropathy

**DOI:** 10.3390/genes11111238

**Published:** 2020-10-22

**Authors:** Olga Mironovich, Elena Dadali, Sergey Malmberg, Tatyana Markova, Oxana Ryzhkova, Aleksander Poliakov

**Affiliations:** 1Federal State Budgetary Institution “Research Centre For Medical Genetics”, Moscow 115478, Russia; genclinic@yandex.ru (E.D.); markova@med-gen.ru (T.M.); ryzhkova@dnalab.ru (O.R.); apol@dnalab.ru (A.P.); 2Federal State Budget Healthcare Institution “Central Children Clinical Hospital” at FMBA of Russia, Moscow 115409, Russia; sergej.malmberg@gmail.com

**Keywords:** hereditary motor neuropathy, *SYT2* mutation, first de novo mutation, electrophysiological testing

## Abstract

Objective: To report the first de novo missense mutation in the *SYT2* gene causing distal hereditary motor neuropathy. Methods: Genetic testing was carried out, including clinical exome sequencing for the proband and Sanger sequencing for the proband and his parents. We described the clinical and electrophysiological features found in the patient. Results: We reported a proband with a new de novo missense mutation, c.917C>T (p.Ser306Leu), in the C2B domain of *SYT2.* The clinical presentation was similar to that of phenotypes described in previous studies. A notable feature in our study was normal electrophysiological testing results of the patient. Conclusions: In this study we reinforced the association between *SYT2* mutations and distal hereditary motor neuropathy. We also described the clinical presentation of the patient carrying this pathogenic variant and provided unusual results of electrophysiological testing. The results showed that a diagnosis of *SYT2*-associated neuropathy should be based on the similarity of clinical manifestations, rather than the results of electrophysiological testing.

## 1. Introduction

Distal hereditary motor neuropathies (dHMNs) comprise a heterogeneous group of rare diseases that share the common feature of predominantly motor, length-dependent neuropathy [1,2]. Previously, a group of authors identified a new type of dHMN caused by autosomal dominant pathogenic variants in synaptotagmin 2 (*SYT2*) [3]. *SYT2* is an integral membrane protein of synaptic vesicles and serves as a calcium sensor for neurotransmitter release, with calcium binding to its C2B domain, activating vesicle fusion [4,5]. This type of disease is associated with presynaptic neuromuscular junction dysfunction and characterized by foot deformities and fatigable ocular and lower limb weakness. In two families that presented with autosomal dominant presynaptic neuromuscular junction disorder and hereditary motor neuropathy, pathogenic variants c.920A>C and c.923C>T were identified [3,6]. These mutations affect adjacent aspartate residues in the C2B calcium-binding domain, Asp307Ala and Pro308Leu. It was shown that mutant *SYT2* is likely to multimerize with endogenous *SYT2* and disrupt its normal properties, resulting in defects in synaptic transmission [3]. In the following study, a family, presenting with clear evidence of both presynaptic neuromuscular transmission impairment and distal motor neuropathy was described. This family carried a pathogenic variant c.1112T>A (p.Ile371Lys), which is also localized in the C2B domain of *SYT2* [4]. Their data indicated that mutant I426K version forms mixed oligomers with WT synaptotagmin 1, thus dramatically reducing neurotransmitter release.

In this study we described a novel de novo variant in the C2B calcium-binding domain of *SYT2,* which caused the same phenotype in the patient, and we presented the results of electrophysiological testing. To identify this variant, clinical exome sequencing was performed for the proband. This method has rapidly become a component of the clinical approach to individuals with rare diseases, including dHMNs [7]. To date, gene panel sequencing is the most common in the diagnosis of dHMNs. Still, many patients and families do not receive a molecular genetic diagnosis after gene panel sequencing because panels are not updated as often and usually do not cover new genes [8]. The advantage of clinical exome sequencing, compared to whole exome sequencing, is the lower cost of the study with approximately equal effectiveness.

## 2. Materials and Methods

Clinical data: The proband was examined at the Research and Counseling Department of the Research Centre for Medical Genetics (RCMG). The patient’s parents provided oral and written consent for this study, approved by the Ethics Committee of RCMG (approval number 1/4 from 15 January 2018). His initial assessment included a full history and physical examination. Electrophysiological studies were performed using a Neurosoft (Neurosoft, Ivanovo, Russia) EMG machine.

Genetic testing: Blood samples from the proband and the unaffected parents were collected, and genomic DNA was extracted using standard methods. Clinical exome sequencing was performed for the proband. The proband’s DNA was analyzed using paired-end reading (2 × 75 bp) on an IlluminaNextSeq 500 sequencer (Illumina, San Diego, CA, USA). Target enrichment was performed with a SeqCap EZ HyperCap Workflow solution capture array, including the coding regions of 6010 genes at that time described as clinically significant in The Human Gene Mutation Database (HGMD Professional, version 2018.1© 2013–2018 QIAGEN, Venlo, The Netherlands). The coding sequence of the *SYT2* gene was completely covered when using this method. Sequencing data were processed using a standard computer-based algorithm from Illumina software, presented on the https://basespace.illumina.com website (Enrichment, version 3.1.0, Illumina, San Diego, CA, USA). Average coverage for this sample was 57.5×, coverage width (10×)–97.72%. The coding sequence of the *SYT2* gene was completely covered when using this method. The sequenced fragments were visualized using Integrative Genomics Viewer (IGV) software (© 2013–2018 Broad Institute, and the Regents of the University of California, USA). Filtering of the variants was based on their frequency—less than 1% in gnomAD—and coding region sequence effect: missense, nonsense, coding indels, and splice sites. The variants’ clinical significance was evaluated according to the guidelines for massive parallel sequencing (MPS) data interpretation [9]. Amplification and Sanger sequencing were performed to validate the exome variant in *SYT2* gene in the proband and its presence in the patient’s parents. Amplifications were carried out by PCR with Taq polymerase in a Veriti Dx Thermal Cycler (Thermo Fisher Scientific, Waltham, MA, USA). The protocol used for amplification included the following steps: 95 °C for 5 min; 32 cycles of 94 °C for 45 s, 62 °C for 45 s, 72 °C for 45 s; 72 °C for 5 min; 4 °C hold. Automatic Sanger sequencing was carried out using an ABIPrism 3100xl Genetic Analyzer (Applied Biosystems, Foster City, CA, USA) according to the manufacturer’s protocol. Sequencing results were analyzed using Chromas (Technelysium Pty Ltd., South Brisbane, Australia). To amplify the fragment encompassing the candidate variants, custom primers were used (according to the NM_177402.4 reference sequence): SYT2_7F: GCACTAGGATGGGTGAGATGAC, SYT2_7R: GACTGGCTCGCTGGTGCCAC.

## 3. Results

### 3.1. Clinical Evaluation

The proband (II-2), a 10-year-old male, was referred to the Research Centre for Medical Genetics because of lower limb weakness, unsteady gait and frequent falling. His parents and siblings were unaffected (Figure 1a). He was born at term from the second pregnancy. Pregnancy and delivery were uneventful. The proband’s parents could not say exactly when the above-mentioned symptoms appeared. His early motor skills were delayed: he was crawling only on his stomach for the first year of life and started walking at the age of 18 months. His cognitive development was considered normal, and he had good academic achievements. General physical examination revealed foot deformities: pes cavus and splayed toes (Figure 1b). He walked by throwing his upper body forward and was unsteady on toes or heels. Neurological examination revealed weakness in proximal and distal lower limb muscles and lower limb wasting. Upper limb strength was normal. All deep tendon reflexes were absent. Sensory examination was normal.

### 3.2. Genetic Analysis

Clinical exome sequencing carried out for the proband revealed a novel heterozygous c.917C>T (p.Ser306Leu, NM_177402.4) variant in exon 7 of the *SYT2* gene (Figure 1c). This variant was not found in the Genome Aggregation Database (gnomAD v.2.1.1) and among samples of 1000 Russian patients’ exomes. Seven different programs (PROVEAN, UMD-predictor, Polyphen2, SIFT, DEOGENE2, MutationTaster, Mutation assessor) predicted this variant to be disease-causing. In addition, the described variant was highly conserved throughout different species (Figure 1d).

We used several tools for a structural/conformational alteration prediction model of the *SYT2* protein with mutation Ser306Leu and comparison with predictions for mutations Asp307Ala and Pro308Leu. Missense3D (http://www.sbg.bio.ic.ac.uk/~missense3d, 2019, Structural Bioinformatics Group, Department of Life Sciences, Imperial College London, UK) predicted the structural changes for Ser306Leu. In particular, this substitution triggered a clash alert (the local clash score for the wild type was 15.99 and the local clash score for the mutant was 41.45) and disrupted all side-chain/side-chain H-bond(s) and/or side-chain/main-chain H-bond(s) formed by buried Ser residue. At the same time, Missense3D detected no structural damage with Asp307Ala and Pro308Leu (Figure 2). Mutfunc (http://www.mutfunc.com, 2019, European Molecular Biology Laboratory, Heidelberg, Germany) predicted protein stability disruption with all three mutations. DynaMut (http://biosig.unimelb.edu.au/dynamut, 2019, Bio21 Institute - University of Melbourne, Melbourne, Australia) predicted *SYT2* protein destabilization with Asp307Ala and Pro308Leu (−1.02 and −0.67 kcal/mol) and stabilization with Ser306Leu.

As a result of Sanger sequencing, it was shown that this variant was absent in proband’s parents (Figure 1c). Consequently, we could conclude the de novo origin of the variant in this family. However, we have not examined the germ cells of the parents and could not rule out germinal mosaicism.

The variant was classified as likely pathogenic according to the guidelines for massive parallel sequencing (MPS) data interpretation (criteria PS2, PM1, PM2, PP3).

### 3.3. Electrophysiological Testing

Motor fiber conduction parameters of the tibial, peroneal, median, and ulnar nerves were determined in a stimulation study. Conduction velocity and residual latency were normal, the amplitude of maximum M-responses was sufficient, and the forms of the responses were not changed. F-wave parameters (latency, chronodispersion of latencies) were normal, there were no blocks of F-waves, “giant” waves, and the proportion of repeated waves did not exceed the norm. No significant changes in the amplitude of the base M-response were determined during a standard test of low-frequency (3 Hz) rhythmic stimulation. The test of rhythmic stimulation with tetanization at low-frequency stimulation did not reveal a decrement of M-response, whereas there was a small increment of the M-response amplitude in the tetanic series (50 Hz 4 s), ranging from 13.9% to 28%, which corresponded to the norm. There were no phenomena of post-tetanus relief and exhaustion. An increase in the amplitudes of the base M-response in early post-tetanic series, as well as immediately after isometric muscle tension of the hand, appeared to be a normal sign of pseudophasylation, which was due to mobilization of acetylcholine resources (Figure 3, Table 1). Thus, no peripheral nerve motor fiber conductive function violations and signs of postsynaptic and presynaptic defects were registered.

In a local-needle study of the anterior tibial muscle on the right side, the pattern of morphofunctional organization of motor units corresponded to the normal pattern, and there were no signs of reinnervation or spontaneous types of activity (Figure 4, Table 2).

## 4. Discussion

Mutations in the *SYT2* gene have previously been associated with a novel autosomal dominant distal motor neuropathy and a presynaptic myasthenic disorder. To date, three families harboring pathogenic variants in *SYT2* have been reported. We are the first to describe a de novo *SYT2* mutation in a patient with dHMN. In this study, we not only reinforced the link between *SYT2* mutations and dHMN, but we also presented unusual electrophysiological testing results.

Synaptotagmin 2 regulates neurotransmitter release at human peripheral motor nerve terminals. *SYT2* is an integral membrane protein of synaptic vesicles and serves as a calcium sensor for neurotransmitter release. We assumed that the de novo c.917C>T mutation caused a serine-to-leucine change at residue 306. Serine residue Ser306 is the first of the five residues that coordinate calcium binding to the C2B domain [3,10]. Previously, it was shown that disruption of these residues leads to loss of calcium binding and dominant-negative disruption of exocytosis in *Drosophila* [11,12]. The critical importance of this domain for protein function, high conservativity of the variant and the de novo origin allowed to characterize this variant as the cause of dHMN development in the patient. In two of the three previously described patients, the causes of the disorder were mutations in adjacent amino acids 307 and 308. These amino acids are located in the same domain, critical for protein function, and are also highly conserved in evolution. We used several tools for a structural/conformational alteration prediction model of the *SYT2* protein with mutation Ser306Leu and compared it with predictions for mutations Asp307Ala and Pro308Leu. We got conflicting results: two of the three programs predicted that the effect of the identified variant differed from that of Asp307Ala and Pro308Leu. However, the results indicated probable pathogenicity of the identified variant.

If we used gene sequencing panel in this case, we would not have received a molecular genetic diagnosis because *SYT2* is not included in most of the currently known panels. Clinical genome sequencing and whole genome sequencing use is advisable in cases where it is impossible to assume a frequent gene responsible for the development of disease. However, a big problem when using these methods is detection of secondary findings, and it requires special attention [13]. All this shows the importance of pre- and post-test genetic counseling of patients with dHMNs.

The clinical signs found in our patient were similar to phenotypes described in previous studies [4,6]. He had a phenotype suggestive of motor neuropathy. The clinical signs included foot deformities, distal weakness, minimal sensory findings, and reduced deep tendon reflexes with evidence of postexercise facilitation. Sensory nerve conduction was normal. In our study, the electrophysiological testing results were notable. Violations of peripheral nerve motor fiber conductive function, signs of postsynaptic and presynaptic defects were not registered, the pattern of morphofunctional organization of motor units corresponded to the norm, and there were no signs of reinnervation or spontaneous activity. Therefore, we noted a feature of neuropathy caused by mutations in the *SYT2* gene—the possible absence of electrophysiological changes in spite of similar clinical presentation. In previous studies, all patients except one, who did not undergo electrophysiological testing, were older than the proband under study. In a 16-year-old patient from family 2 (III.6), CMAP amplitudes were normal, but he had a milder clinical picture: he was asymptomatic, although general examination revealed pes cavus, bilateral clawing of his toes, and mild wasting of distal lower limb muscles. In this regard, it was logical to assume the absence of electrophysiological changes was related to the young age of our patient. However, in the study by Hermann D. with co-authors, mutations at Asp307Ala and Pro308Leu have been associated with nonprogressive motor neuropathy. In the next study, however, the authors pointed to slow progression of the disease in affected family members carrying the mutation Ile371Lys. Perhaps this was due to the fact that the effect of mutation Ile371Lys differs from that of mutations Asp307Ala and Pro308Leu. In relation to the above, we could not rule out that electrophysiological changes have been registered in our proband in the last few years; however, this would require further examination.

In this work we implied that when making a diagnosis of *SYT2*-associated neuropathy, we need to rely more on the similarity of clinical manifestations, rather than the results of electrophysiological testing.

## 5. Conclusions

We described a new case of *SYT2*-associated neuropathy that will lead to better understanding of this pathology in the future.

## Figures and Tables

**Figure 1 genes-11-01238-f001:**
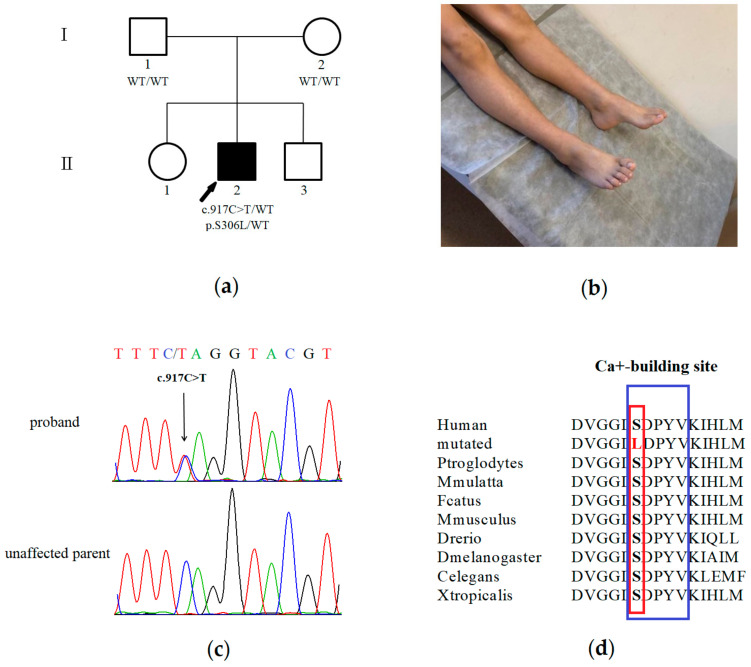
Clinical presentation and genetic testing of the proband with a variant in the *SYT2* gene: (**a**) The family tree showing the affected proband and unaffected parents and siblings; (**b**) Proband showing foot deformity and lower limb wasting; (**c**) Sanger sequencing results demonstrating the heterozygous c.917C>T missense *SYT2* mutation in the proband; (**d**) Position of the c.917C>T mutation (red) in the *SYT2* protein structure. This variant was highly conserved throughout different species. The cytoplasmic C2B domain of *SYT2* is highlighted in blue.

**Figure 2 genes-11-01238-f002:**
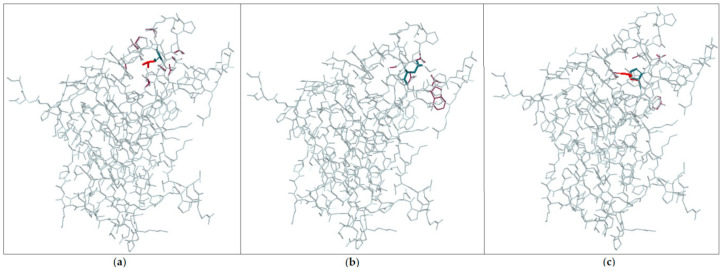
Prediction model of the *SYT2* protein: (**a**) with mutation Ser306Leu; (**b**) with mutation Asp307Ala; (**c**) with mutation Pro308Leu; wild type is highlighted with gray and blue; mutant is highlighted with red and dark red.

**Figure 3 genes-11-01238-f003:**
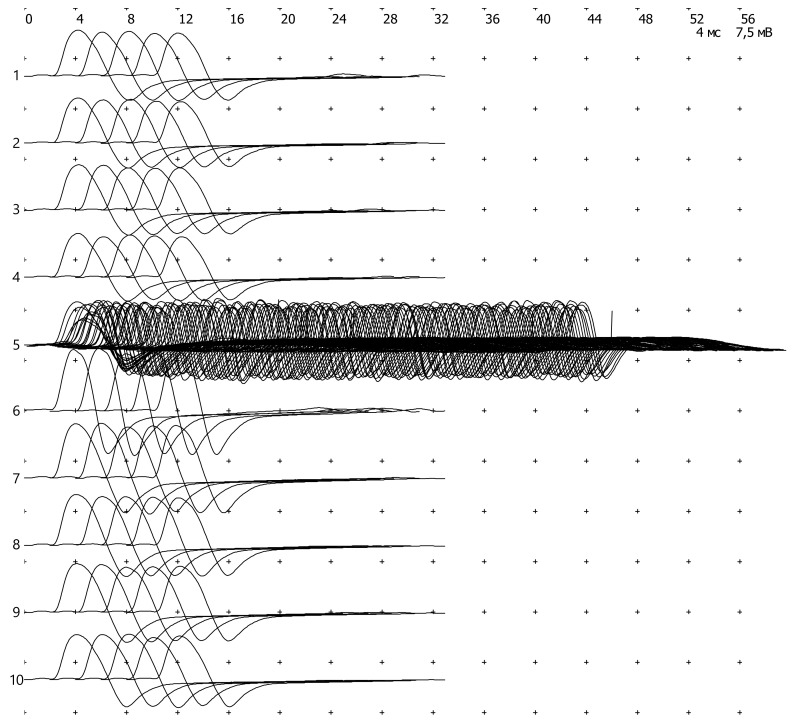
Test of rhythmic stimulation with tetanization (m. adductor pollicis ulnaris sin, C8 T1, 20 mA, duration—200 μs).

**Figure 4 genes-11-01238-f004:**
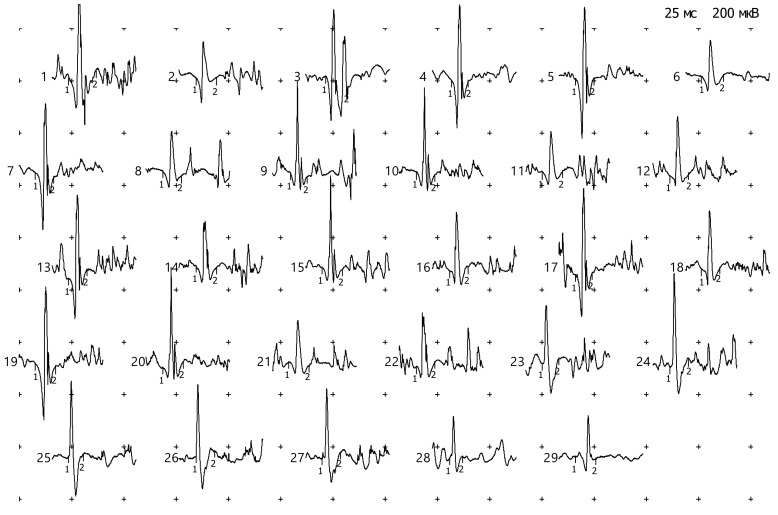
Motor unit potentials (MUPs) (m. tibialis anterior dex, peroneus, L4 L5 s1).

**Table 1 genes-11-01238-t001:** Test of rhythmic stimulation with tetanization (m. adductor pollicis ulnaris sin, C8 T1, 20 mA, duration—200 μs).

N	Frequency, Hz	Number of Incentives	Amplitude M-Response, mV	Decrement., % (1–5)	Decrement., % (1–last)	Area, mV × ms	Decrement of Area, % (1–last)
1	3.0	5	10.5	4.5	4.5	28.1	5.4
2	3.0	5	10.4	7.4	7.4	27.7	7.6
3	3.0	5	10.5	6.1	6.1	27.7	4.4
4	3.0	5	10.1	6.6	6.6	26.7	8.5
5	50.0	200	9.96	2.3	+13.9	26.1	24.2
6	3.0	5	15.5	0.8	0.8	34.3	+5.4
7	3.0	5	13.3	2.0	2.0	34.1	1.4
8	3.0	5	12.2	5.2	5.2	32.2	6.6
9	3.0	5	11.7	5.2	5.2	31.1	5.6
10	3.0	5	10.7	2.1	2.1	28.3	3.4

**Table 2 genes-11-01238-t002:** Motor unit potentials (MUPs) (m. tibialis anterior dex, peroneus, L4 L5 s1).

Duration of MUPs						
**Min. Dur.,** **ms**	**Max. Dur.,** **ms**	**Average Dur.,** **ms**	**Norm.,** **ms**	**Dur. Deviation,** **%**	**Stage**	**Number of MUPs**
6.52	12.7	9.19	9.2	Norm	Norm	29
**Amplitude of MUPs**						
**Min. Amp,** **mkV**	**Max. Amp, mkV**	**Average Amp.,** **mkV**	**Norm Amp., mkV**	**Amp. Deviation,** **%**	**Amp.** **> 1 mV**, **%**	**Polyphase MUPs,** **%**
194	517	355	475	Norm	0	5.8

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
