# Peer review of "Identification of a Novel de Novo Variant in the SYT2 Gene Causing a Rare Type of Distal Hereditary Motor Neuropathy"

_genes, 2020, doi:10.3390/genes11111238_

Round 1

Reviewer 1 Report

This manuscript by Mironovich et al, provides an interesting finding of a novel mutant variant of calcium sensor protein SYT2 in young male individual possibly in a Russian family. Authors investigated the clinical linkage of this mutant variant with hereditary distal motor neuropathies, a rare kind of neurological disorder. Given that there are a very few research investigations have been conducted on this important gene pathogenic mutations, the present study provides some important clues to this genetic disorder. However, the manuscript needs to be improved quite a bit . Please find the suggestions below:

1) In Abstract section, Line 21, the meaning of the sentence is ambiguous, please rewrite it in a meaningful way.

2) The Introduction section is too much brief and lacks important background information and citations, please improve it including the info of other mutant variants of this gene and their so far known involvement in motor neuropathy.

3) In Materials & Methods section, Line 39, what is meant by " local ethics committee"? It should be some specific institutional ethics committee.

4) The methodology of targeted sequencing of mutant variant has been poorly described, please include the specific technical details.

5) Regarding the SYT2 mutation, it seems from previous studies that the amino acid residues at 306, 307 and 308 are prone to mutation. While 307 and 308 have been already described as the congenital ones, it is not clear how could the authors claim that 306 mutation is not congenital rather a genetically inherited one.

6) Please provide a structural/conformational alteration prediction model of SYT2 protein with mutation at 306, and whether this predicted model is similar to mutations at 307 and 308. This data will help strengthen the claim for pathogenic nature of this mutation.

7) Given that the mutations at 307 and 308 have been reported to be associated with non-progressive motor neuropathy, it is not clear how the authors could arrive to the conclusion that mutation at 306 would cause progressive neuropathy as they mentioned in Page 6, Line 144.

8) Citations and References need to formatted well.

Author Response

Dear reviewer! Thank you very much for the thorough analysis of our article. I think all the corrections will benefit the article! All changes are highlighted in blue. The corrections requested by another reviewer are highlighted in yellow. We have also corrected the type of article on Case Report. Maybe this information will be important to you.

  1. We have rewritten this sentence and divided it into two parts (line 20-22). We hope it has become clearer.
  2. We have included the info of other mutant variants of this gene: their effect on SYT2 protein and processes that are disrupted by these mutations (line 32-46).
  3. Yes, it is the ethics committee of Research Centre for Medical Genetics. We included this information (line 58).
  4. We have included the specific technical details about Sanger sequencing (line 76-82). In addition we decided to add information about exome sequencing (line 63-70)
  5. In part 3.2 we clarified that this variant was absent in the parents of the proband and we can conclude about the de novo origin of the variant in this family. Also we added that we cannot rule out germinal mosaicism (line 116-119).
  6. We used several tools for a structural alteration prediction model of SYT2 protein. Add these date in Results and Discussion(line 107-115, line 177-182).
  7. You are right. We have corrected this paragraph so that this thought is a theory. We also added facts for and against this (line 196-209).
  8. We tried to fix all mistakes.

Reviewer 2 Report

The obtained results could be interesting but the drafting of the manuscript must be absolutely modified. Since only one case is described, the paper should be developed as a case report. The introduction must be extensively developed by describing the role of the clinical exome in daily laboratory activities. It is essential to introduce the importance of pre and post test genetic counseling and this topic must be included in the introduction and in discussions.

Author Response

Dear reviewer! Thank you very much for the analysis of our article. We have corrected the type of article on Case Report. We have also added describing the role of the clinical exome in daily laboratory activities (line 48-54) and the importance of pre and post test genetic counseling (line 183-188). All changes are highlighted in blue and yellow.

Round 2

Reviewer 1 Report

Thanks to the authors for their efforts to significantly improve the scientific quality of the manuscript. However, the manuscript requires thorough revision for grammar and language editing especially in some instances, e.g. in Abstract, line 20 please correct the spelling of "assosiation", and please remove the word "interesting" in Line 22.

Also, I would like to ask the authors to add one more info in Materials and Methods section before the final acceptance of the manuscript:

Please include the Thermal cycling profile for PCR amplification of SYT2 fragment using the custom primers. Honestly, I am not very happy with the design and selection of this primer pair as it may form multiple intra-sequence hybridizations. I hope authors have considered all possible parameters to select this primer pair as the most suitable one among other options.

Thanks

Author Response

Dear reviewer! We tried to edit grammar and language. We have also added the thermal cycling profile for PCR amplification (line 79-80). Of course, we have considered all possible parameters of primer pair and have chosen the most suitable one. According to BLAT and BLAST both primers are unique. In addition the correct selection is confirmed by the results of sequencing. If necessary we can send you sequencing tracks.

Thank you again for the analysis of our article! I think all the corrections will benefit the article!
